# Regulation of Gut Microbiota and Metabolic Endotoxemia with Dietary Factors

**DOI:** 10.3390/nu11102277

**Published:** 2019-09-23

**Authors:** Nobuo Fuke, Naoto Nagata, Hiroyuki Suganuma, Tsuguhito Ota

**Affiliations:** 1Department of Nature & Wellness Research, Innovation Division, KAGOME CO., LTD., Nasushiobara 329-2762, Japan; Nobuo_Fuke@kagome.co.jp (N.F.); Hiroyuki_Suganuma@kagome.co.jp (H.S.); 2Department of Cellular and Molecular Function Analysis, Kanazawa University Graduate School of Medical Science, Kanazawa 920-8640, Japan; nnagata@staff.kanazawa-u.ac.jp; 3Division of Metabolism and Biosystemic Science, Department of Medicine, Asahikawa Medical University, Asahikawa 078-8510, Japan

**Keywords:** metabolic endotoxemia, lipopolysaccharide, gut microbiota, dietary factors

## Abstract

Metabolic endotoxemia is a condition in which blood lipopolysaccharide (LPS) levels are elevated, regardless of the presence of obvious infection. It has been suggested to lead to chronic inflammation-related diseases such as obesity, type 2 diabetes mellitus, non-alcoholic fatty liver disease (NAFLD), pancreatitis, amyotrophic lateral sclerosis, and Alzheimer’s disease. In addition, it has attracted attention as a target for the prevention and treatment of these chronic diseases. As metabolic endotoxemia was first reported in mice that were fed a high-fat diet, research regarding its relationship with diets has been actively conducted in humans and animals. In this review, we summarize the relationship between fat intake and induction of metabolic endotoxemia, focusing on gut dysbiosis and the influx, kinetics, and metabolism of LPS. We also summarize the recent findings about dietary factors that attenuate metabolic endotoxemia, focusing on the regulation of gut microbiota. We hope that in the future, control of metabolic endotoxemia using dietary factors will help maintain human health.

## 1. Introduction

Lipopolysaccharide (LPS) is a component of the outer membrane of gram-negative bacteria and is known to induce a variety of inflammatory reactions through Toll-like receptor 4 (TLR4). Injection of LPS into human blood elicits an inflammatory response [1,2], but it was thought that LPS is rarely detected in human blood, except under pathological conditions such as infection and colitis. However, in 2007, Cani et al. showed that mice fed with a high-fat diet had higher blood LPS levels than normal chow-fed mice, resulting in inflammation of the liver and adipose tissue, which led to the development of NAFLD and insulin resistance, and the authors defined this condition as metabolic endotoxemia [3]. Since then, studies on metabolic endotoxemia have been conducted for a variety of diseases. It has been reported that blood LPS levels are higher in humans with obesity [4], type 2 diabetes [5], NAFLD [6], pancreatitis [7], amyotrophic lateral sclerosis [8], and Alzheimer’s disease [8] than those in healthy individuals. Although the causal relationship between metabolic endotoxemia and disease onset is unclear, it is expected to be an interesting target in the future from the viewpoint of disease prevention and treatment. In recent years, the association between metabolic endotoxemia and dietary factors, and the mechanism by which fat intake induces metabolic endotoxemia have been actively studied. In contrast, dietary factors that suppress metabolic endotoxemia have also been explored. Here, we review the relationship between fat intake and induction of metabolic endotoxemia, focusing on gut dysbiosis and the influx, kinetics, and metabolism of LPS. We also summarize the recent findings in humans and animals about dietary factors that attenuate metabolic endotoxemia, focusing on regulation of gut microbiota.

## 2. Fat Intake and Metabolic Endotoxemia

### 2.1. Dysbiosis

As Cani et al. reported an increase in blood LPS levels due to a high-fat diet in mice, the mechanism of LPS influx by fat ingestion has been investigated. LPS content both in cecal contents and blood was concomitantly increased by fat ingestion [9], and this increase of LPS was suppressed with oral administration of intestinal alkaline phosphatase, a LPS inactivating enzyme [9]. Oral administration of ampicillin and neomycin, broad-spectrum antibiotics that are poorly absorbed, also suppressed the increase in blood LPS concentration induced by a high-fat diet [10]. These reports suggest that intestinal bacteria are an important source of LPS. In particular, Cani et al. demonstrated changes in intestinal flora (reduction in *Bacteroides*, *Bifidobacterium*, and *Eubacterium*) due to a high-fat diet. Thus, dysbiosis of the intestinal flora due to a high-fat diet has attracted attention as a possible cause for metabolic endotoxemia. Changes in the intestinal bacteria due to ingestion of a high-fat diet have been studied in animals and humans and have been summarized in a review by Netto Candido et al. [11]. In animals, it has been reported that a high-fat diet increases the proportion of *Firmicutes*, *Proteobacteria*, and the ratio of *Firmicutes* to *Bacteroidetes*. On the other hand, in humans, it has been reported that high-fat dietary intake increases the proportion of *Bacteroidetes* and decreases the proportion of *Firmicutes* and *Proteobacteria*. One possible cause of the different changes in the gut microbiota at the phylum level (e.g., *Firmicutes*, *Bacteroidetes*, *Proteobacteria*) in human and animal studies is the difference in the type of fat consumed. The high-fat diet used in animal experiments (e.g., Research Diets Inc., catalog# D12451) contains lard, while human studies assess fat intake in daily diets. Devkota et al. evaluated the gut microbiota in C57BL/6 mice fed a low-fat diet, a high-fat diet with lard, or a high-fat diet with milk fat for 21 days [12]. In this experiment, both high-fat diets were isocaloric, rich in saturated fatty acids, and 37% of the ingested kcal were from fat. As a result, the proportion of *Firmicutes* increased and that of *Bacteroidetes* decreased in the gut microbiota of mice fed a high-fat diet containing lard, compared to mice fed a low-fat diet. In contrast, in mice fed a high-fat diet containing milk fat, the proportion of *Firmicutes* decreased and that of *Bacteroidetes* increased compared to the low-fat diet fed mice. Interestingly, Devkota et al. also identified specific bacteria that increased only by ingestion of a high-fat diet containing milk fat [12]. Compared to mice fed a low-fat diet, or a high-fat diet containing lard, mice fed with a high-fat diet containing milk fat had increased proportions of *Bilophila wadsworthia*, a sulfite-reducing bacterium, in gut microbiota. They also elucidated the mechanism underlying this increase; intake of milk fat increased the level of taurocholic acid in bile. *Bilophila wadsworthia* populations increased by utilizing sulfur components in taurocholic acid, causing intestinal inflammation in mice. An increase in total fecal bile acid and a concomitant increase in *Bilophila wadsworthia* in the gut microbiota was also reported in humans upon dietary intake of animal fat [13]. Natividad et al. also showed that increased *Bilophila wadsworthia* in mice fed a high-fat diet contributed to increased blood LPS levels (they measured soluble CD14 as a surrogate marker), increased fasting blood glucose levels, and the development of a fatty liver [14]. As *Helicobacter pylori* was discovered as a pathogen in gastric cancer, some pathobionts may also exist for induction of metabolic endotoxemia (however, this cannot be detected by evaluating changes of the gut flora at the phylum levels). We further discuss the bacterial genera that are thought to be associated with metabolic endotoxemia in Section 4. It is also necessary to consider dietary LPS as a source of LPS. For example, milk has been reported to contain high concentrations of LPS in some commercial products [15]. Multiple animal studies have reported that ingested LPS may contribute to increased blood LPS levels. Specifically, Kaliannan et al. measured blood LPS levels 45 min after ingestion of LPS alone or corn oil and LPS in mice [9]. It showed that blood LPS levels were elevated when corn oil and LPS were co-administered. Lindenberg et al. reported that LPS concentrations in the blood were higher in mice fed a high-fat diet containing LPS than in mice fed a high-fat diet without LPS [16]. However, the effect of LPS levels in food on blood LPS levels has not been adequately studied in humans and further studies are needed.

### 2.2. Mechanisms of the Influx of LPS into the Bloodstream

The gut is protected by a barrier consisting of a mucin layer and epithelial cells. Thus, even if the number of gram-negative bacteria that produce LPS increases in the gut, it is unlikely that the bacterium itself will invade the body. The limulus amebocyte lysate assay used to measure LPS recognizes lipid A, a glycolipid moiety of LPS [17], but because lipid A is embedded in the outer membrane of gram-negative bacteria [18], elevated blood LPS levels suggest that LPS released from gram-negative bacteria is flowing into the blood. In an *in vitro* study with *Escherichia coli*, the concentration of free LPS in the culture medium increased with bacterial growth, but the addition of antibiotics stimulated further LPS release [19]. In addition, Jin et al. suggested that treatment with penicillin and erythromycin killed the gram-negative bacteria, *Bacteroides* and γ-*Proteobacteria*, leading to increased blood LPS levels in mice [20]. Radilla-Vázquez et al. conducted a correlation analysis of blood LPS levels with fecal *Escherichia coli*, *Prevotella*, and *Bacteroides fragilis* counts in humans and reported that the lower the number of gram-negative bacteria *Escherichia coli*, the higher the risk of increased blood LPS levels [21]. These reports suggest that LPS release by lysis as well as the increase in gram-negative bacteria may be important factors in increasing blood LPS levels, which may contribute to the inconsistent relationship between changes in intestinal flora and blood LPS levels described above.

With respect to the influx of free LPS, Laugerette et al. reported that in an *in vitro* assay system using the intestinal epithelial cell line caco-2, LPS permeability to the basal side was increased in the presence of oleic acid, 2-oleoylglycerol, soybean lecithin, cholesterol, and sodium taurocholate [22]. In addition, Clement-Postigo et al. reported a positive correlation between increased LPS levels in the chylomicron fraction and increased triglyceride concentration in serum up to 3 h after a high-fat meal [23]. LPS uptake in chylomicrons has been observed by immunoelectron microscopy [22]. These results suggest that released-LPS in the intestine is taken up into micelles during lipid absorption, and then LPS is absorbed from the intestine together with lipids. In mice, ingestion of a high-fat diet has been reported to increase intestinal permeability by inhibiting the mRNA expression of tight junction-related factors, zonula occludens-1 (ZO-1) and occludin in intestinal epithelial cells [10]. This increase in intestinal permeability is markedly inhibited by antibiotic administration [10], suggesting that it is not the direct effect of lipids but rather a change in intestinal flora. Indeed, secondary bile acids metabolized by enteric bacteria are known to inhibit expression of intestinal tight junction proteins [24,25]. Increased intestinal LPS has been reported to destroy the tight junction of intestinal epithelial cells through TLR4 [26]. Although ingestion of a high-fat diet broadly enhances intestinal and colonic permeability [27], permeability in the colon is closely related to increased blood LPS levels [28,29]. Therefore, disruption of the barrier function by a high-fat diet may have also contributed to the LPS inflow, and the colon may be important as a site of the absorption. The transit time of colonic contents is also probably important. In mice, Anitha et al. suggested that saturated fatty acids induced apoptosis of neurons in the large intestine, reduced peristalsis, induced constipation, and increased blood LPS levels [30]. On the other hand, Reichardt et al. similarly evaluated peristalsis of the large intestine by ingestion of a high-fat diet, but did not observe a clear decrease in peristalsis and an increase in blood LPS levels [31]. Anitha et al. and Reichardt et al. used high-fat diets where either 60% or 30%, respectively of ingested kcal came from fat. Although the ratio of fat to energy intake varied, it has been reported that blood LPS levels increased by consumption of a high-fat diet with 30% of kcal ingested being from fat [32,33]. Therefore, the reason for the lack of increase in blood LPS levels in the study of Reichardt et al. is not considered to be a difference in the fat content of the diet. Ingestion of a high-fat diet does not simply increase blood LPS levels, and retention time of colonic contents due to constipation may also contribute to absorption of LPS.

### 2.3. Kinetics and Activity of LPS

The LPS concentration in the portal blood is approximately 10 times higher than the LPS concentration in the peripheral blood [34], suggesting that a part of the LPS released in the intestinal tract is flowing from the portal vein. On the other hand, LPS which is concomitantly absorbed with lipids binds to lipoproteins in chylomicrons via LPS-binding protein (LBP) [35], and is thought to pass through the lymphatic system, flow into the blood stream from the left subclavian vein, and then circulate throughout the body. It is reported that blood LPS is bound to various lipoproteins, with plasma LPS concentrations of 31%, 30%, 29%, and 10% for the very low-density lipoprotein (VLDL) fraction, low-density lipoprotein (LDL) fraction, high-density lipoprotein (HDL) fraction, and free LPS, respectively [36]. In addition, LPS bound to lipoproteins of HDL has been reported to be transferred to VLDL and LDL by LBP and phospholipid transfer protein [37], suggesting that the LPS concentration of each lipoprotein fraction changes actively. There are several reports that bioactivity of LPS bound to lipoprotein varies with the type of lipoprotein. First, Vreugdenhil et al. evaluated the effect of chylomicrons, HDL, LDL, and VLDL on the production of tumor necrosis factor-α (TNF-α) from human peripheral blood mononuclear cells on LPS stimulation and showed that chylomicrons inhibited TNF-α production the most [35]. Emansipator et al. reported that a mix of LPS with LDL or HDL decreased the spike recovery of LPS activity in the limulus amebocyte lysate test, and that incubation of LPS with apo A1 decreased the febrile response of rabbits when injected compared to those without apo A1 [38]. In a study using human mononuclear cells [39] and the mouse macrophage cell line Raw 264.7 [40], it was reported that LPS bound to HDL showed reduced interleukin-6 (IL-6) and TNF-α production. VLDL has also been reported to inhibit LPS-induced activation of nuclear factor κB (NF-κB) [41]. On the other hand, oxidized LDL has been shown to promote NFκB activation with LPS in macrophages [42], suggesting that binding to lipoproteins not only decreases LPS activity but also may promote inflammatory responses.

Increased LPS content has been reported in the livers of mice fed a high-fat diet [43], suggesting that the liver is an important site for LPS clearance. Ninety percent of the free LPS that entered the bloodstream is captured by liver resident macrophages (i.e., Kupffer cells) within 1 h [44]. LPS bound to HDL attaches primarily to sinusoidal epithelial cells of the liver [40,44], but it shows slower blood kinetics than free LPS, with 50% present in plasma even 1 h after administration and the amount accumulated in the liver accounted for only 15% of the dose [44]. LPS bound to HDL on the other hand is distributed widely to organs other than the liver, such as the kidney and adipose tissue [44]. LPS accumulated in the liver is inactivated by acyloxyacyl hydroxylase produced by Kupffer cells regardless of free or HDL-bound form [44]. Previously, in a mouse model of high-fat diet plus streptozotocin-induced non-alcoholic steatohepatitis-hepatocellular carcinoma, fecal LPS levels were continuously elevated from six weeks, while liver LPS levels were transiently elevated at eight weeks, followed by increased plasma LPS levels [45]. This report suggests that the liver acts as the first barrier against LPS entering from the intestinal tract and that liver dysfunction leads to elevated blood LPS levels. Interestingly, LPS administration in mice increased the expression of apolipoprotein AIV in the liver via TLR4, suggesting that the liver has a mechanism to increase HDL production and protect itself against LPS stimulation [46].

## 3. Dietary Factors that Decrease Blood LPS Levels

Previous reports investigating the effects of dietary factors on blood LPS levels are summarized in Table 1 (human interventional studies), Table 2 (human epidemiological studies) and Table 3 (animal studies). The findings about representative food categories are reviewed in the following sections.

### 3.1. Probiotics

Probiotics were defined by Fuller in 1989 as “A live microbial feed supplement which beneficially affects the host animal by improving its intestinal microbial balance” [47], and has been studied mainly for lactic acid bacteria and bifidobacteria. Since metabolic endotoxemia has been implicated in gut dysbiosis, the effects of probiotics have been investigated. However, the results in humans are unfavorable (Table 1). Lever et al. administered 195 mL of Yakult light (containing 2 × 10^10^ colony-forming unit (CFU) of *Lactobacillus casei* Shirota) for three months to individuals with metabolic syndrome. The absence of detectable blood LPS in this study led to an assessment of the surrogate LBP level, which was significantly higher in the Yakult light-fed group than in the non-fed group [48]. Pei et al. conducted a nine-week study in which low-fat yogurt was ingested in healthy or obese individuals, however no significant decrease in blood LPS or LBP levels was observed [49]. In addition, Pei et al. studied whether low-fat yogurt could be administered before a meal to suppress the increase in blood LPS after a meal [50] and found no efficacy. On the other hand, there have been several reports of the efficacy of probiotics in animal studies (Table 3) [43,51,52,53,54,55]. *Lactobacillus rhamnosus*, *Lactobacillus sakei*, *Lactobacillus acidophilus*, *Lactobacillus plantarum*, *Bifidobacterium longum*, *Bifidobacterium infantis*, and *Bacillus cereus* are used as species, and the dosage ranges from 10^7^ to 10^10^ CFU/day for four to twelve weeks. These animal studies used a high-fat diet, a high-fat high-sucrose diet, or a Zucker-Lep*^fa^*^/*fa*^ obesity model. In addition to a significant decrease in blood LPS or LBP levels, improvement of obesity, glucose metabolism, and dyslipidemia was also observed. Since the effects of probiotics are strain-specific, it is expected that the effects of strains that have been effective in animal studies will be verified in humans.

### 3.2. Prebiotics

Prebiotics was defined by Gibson and Roberfroid in 1995 as “nondigestible food ingredients that beneficially affect the host by selectively stimulating the growth and/or activity of one or a limited number of bacterial species already resident in the colon, and thus attempt to improve host health” [56], and among the food components, dietary fiber and oligosaccharides are known as typical prebiotics. To date, human intervention studies have been conducted with oligofructose [57], inulin [58,59], galacto-oligosaccharides [60,61,62], resistant dextrin [63], insoluble dietary fiber [64], and whole grains (Table 1) [65]. Oligofructose is an oligosaccharide containing one molecule of glucose and several molecules of fructose and is found in many fruits and vegetables. Inulin is a type of fructose-polymerized polysaccharide that is abundant in vegetables such as burdock and onion. In intervention studies with oligofructose [57] and inulin [58,59], subjects with obesity, overweight subjects, and subjects with type 2 diabetes consumed 10–21 g of test substances for 8–12 weeks. Two of the three studies showed a significant decrease in blood LPS levels [57,58]. One study also showed a decrease in plasminogen activator inhibitor-1 (PAI-1), a risk indicator for thrombosis [57], and the other study showed an improvement in glucose metabolism [58]. Galacto-oligosaccharides are oligosaccharides in which multiple molecules of galactose are attached to one molecule of glucose. Similar to oligofructose, there have been three reports of interventional trials for galacto-oligosaccharide in obese, overweight, and type 2 diabetic patients. One study showed that galacto-oligosaccharides reduced blood LPS levels, and improved obesity by suppressing appetite [62]. In mice, chronic administration of LPS has been reported to induce hyperphagia by decreasing leptin sensitivity of afferent vagal nerves [66], and the reduced blood LPS levels and appetite suppression seen with galacto-oligosaccharide administration are of interest in supporting an association between LPS and appetite.

### 3.3. Polyphenols

Polyphenols are secondary metabolites found in plants and are responsible for protection against oxidative stress, UV damage, and pathogenic microorganisms [67]. Polyphenols are found in a wide range of foods, including vegetables, fruits, tea, beans, and spices, and their consumption has been reported to improve metabolic syndrome (decreased body weight, decreased blood pressure, improved glucose metabolism, and improved lipid metabolism) [68]. However, up to 27% of ingested polyphenols are detected in urine [69], suggesting that many of them are not absorbed and reach the large intestine [70]. Since polyphenols reaching the large intestine have been reported to alter the proportions of microbiota [71], it is expected that the effect of polyphenols against metabolic syndrome is mediated through the improvement of dysbiosis and of the accompanying metabolic endotoxemia. There are two human intervention studies investigating the relationship between polyphenol intake and blood LPS, both of which evaluated the inhibitory effect on postprandial elevation of blood LPS levels (Table 1) [72,73]. In the study performed by Ghanim et al., healthy individuals ingested capsules containing 100 mg of resveratrol and 75 mg of polyphenol 10 min before a 930-kcal high-fat, high-carbohydrate meal. Blood LBP levels up to 5 h after a meal were evaluated and showed increased blood LBP levels in the placebo group but not in the capsule group [72]. On the other hand, Clemente-Postigo et al. administered 272 mL of red wine to humans simultaneously with excessive fat and found no effect on either blood LPS or LBP levels [73]. The efficacy of polyphenols has been also reported in animal studies. The effects of grape seed proanthocyanin [29,33], resveratrol [74], apple-derived polymeric procyanidins [75], genistein [76], isoflavone [77], and syringarecinol [78] on blood LPS levels in animal models have been reported (Table 3). In particular, L’openz et al. reported that six-month administration of genistein to high-fat diet-fed mice reduced their blood LPS levels and improved their spatial memory ability [76]. Cho et al. administered syringalesinol to 40-week-old mice for 10 weeks and showed that the decrease in blood LBP levels was accompanied with suppression of changes in immune cells due to aging (decreased naive T cells and decreased T-cell proliferation) [78]. It has also been reported that adoption of a high-fat diet results in abnormal differentiation of bone marrow hematopoietic stem cells due to increased blood LPS levels [79], suggesting that the effect of syringalecinol on immunoaging might be also exerted in other models of metabolic endotoxemia.

### 3.4. Sulfated Polysaccharide

Sulfated polysaccharides are widely present in animal tissues and seaweed and are used industrially as anticoagulants, pharmaceuticals, and gelling agents for foods. The effect of sulfated polysaccharides on metabolic endotoxemia has been studied only in animals (Table 3). Intervention studies with sea cucumber-derived sulfated polysaccharides [80,81], acaudina molpadioides-derived fucosylated chondroitin sulfate [82], chicken-derived chondroitin sulfate [83] or fucoidan [84] have been performed. Of these studies, two showed that administration of sulfated polysaccharides to high-fat diet-fed mice increased the amount of short-chain fatty acids in the intestinal tract, decreased the blood LPS or LBP concentration and attenuated weight gain [80,82]. Zhu et al. also reported the same effect of sulfated polysaccharides in chow-fed lean mice [81]. Liu et al. demonstrated that exhaustive exercise with a treadmill significantly impaired kidney function, decreased fecal butyrate levels, changed intestinal morphology, and induced metabolic endotoxemia [83]. Their study is interesting in showing that exercise stress also increased blood LPS levels, and that dietary factors are also effective in the model mice.

### 3.5. Other Dietary Components/Extracts/Foods

In the study by Abboud et al., obese or over weight subjects ingested 30 g of glutamine per day for eight weeks (Table 1) [85]. As a result, their blood LPS levels and waist circumference decreased. In an epidemiological study conducted with healthy subjects, 25-hydroxy vitamin D was reported to negatively correlate to blood LPS levels (Table 2) [86]. The protective effect of vitamin D is supported by animal studies in which vitamin D-deficient mice, exposed to a bacterial pathogen, exhibited lower LPS detoxification activity of the intestine and greater endotoxin translocation [87]. The effect of other dietary components, including tetrahydro iso-alpha acid [88], rhein [89], phlorizin [90], capsaicin [91], rutin [92], and lycopene [93] on blood LPS levels in animals has also been reported (Table 3). Among them, administration of tetrahydro iso-alpha acid [88], phlorizin [90], or rutin [92] to high-fat diet-fed mice or *db*/*db* mice improved metabolic impairment. Administration of rhein [89], or lycopene [93] to high-fat diet-fed mice showed a unique effect; they not only reduced blood LPS levels but also prevented high-fat diet-induced memory impairment. Kang et al. showed that administration of antibiotics to mice given capsaicin abolished the effect of capsaicin on blood LPS levels [91]. They also showed that capsaicin-induced protection against high-fat diet-induced blood LPS increase is transferrable by fecal microbiota transplantation.

It has also been reported that intervention with crude food extracts or the food itself can lower blood LPS levels in animals (Table 3). We studied the effect of broccoli sprout extract, enriched in functional glucosinolate “glucoraphanin” (details are described in Section 4) [94]. Anhê et al. examined the effects of extracts from cranberry [95] or camu camu [96]. Camu camu is an Amazonian fruit that contains an abundance of vitamin C and flavonoids such as ellagic acid, ellagitannins, and proanthocyanidins. Administration of camu camu extracts to high-fat/high-sucrose diet-fed mice reduced plasma bile acid pool size, altered gut microbiota composition, and reduced blood LPS levels. Dey et al. reported that administration of green tea extract to high-fat diet-fed mice suppressed inflammation and gut permeability especially in the ileum and colon, and reduced LPS influx from the portal vein [34]. The reduction of blood LPS levels by feeding with Tartary buckwheat protein was reported by Zhou et al. [32]. This study is valuable in that it elucidates one of the underlying mechanisms by which plant protein intake leads to improvement of metabolic abnormalities. Intervention studies with cocoa [97], nopal [98], and steamed fish meat [99] have been performed. Among these, Zhang et al. performed unique experiments [99]. They divided mice into four groups, and fed them *ad libitum* with normal chow, steamed fish, pork or beef at 9:00 and 18:00 daily for eight weeks. As a result, only mice group fed with steamed fish showed decreased blood LBP levels compared to the other three groups.

### 3.6. Chinese Medicines

The effect of the Chinese medicines; geniposide + chlorogenic acid [100], potentilla discolor bunge water extract [101], ganoderma lucidum mycelium water extract [102], semen hoveniae extract [103], and shenling baizhu powder [104] on blood LPS levels have been reported in animals (Table 3). The combination of geniposide and chlorogenic acid is included in a traditional Chinese medicine, Qushi Huayu Decoction. Peng et al. indicated that administration of geniposide and chlorogenic acid to high-fat diet-fed mice restored colonic tight junctions by inhibiting down-regulation of RhoA/Rho-associated kinase signaling, and reduced blood LPS levels and hepatic LBP protein levels [100]. Han et al. examined the effect of potentilla discolor bunge water extract in type 2 diabetic mice induced by high-fat diet feeding and streptozotocin injection [101]. The results showed that fecal LPS levels in the type 2 diabetic model mice were significantly increased compared to the control normal mice. The administration of potentilla discolor bunge water extract to mice reduced fecal LPS levels, decreased blood LPS levels and increased the expression levels of tight junction proteins (Claudin-3, ZO-1, and Occludin) in the colon. Chang et al. studied the effect of ganoderma lucidum mycelium, a *Basidiomycete* fungus [102]. They showed the dose-dependent effect of ganoderma lucidum mycelium water extract on blood LPS reduction, suggesting that high molecular weight polysaccharides (>300 kDa) isolated from the extract is an effective component. Ping et al. reported that the extract of semen hoveniae, a seed of Hovenia dulcis Thunb rich in dihydromyricetin and quercetin, decreased blood LPS levels in a mouse model of alcohol-induced liver injury [103]. It has been reported that administration of shenling baizhu, a mixture of ten different traditional Chinese medicinal herbs, to high-fat diet-fed mice decreased LPS levels in the portal vein [104].

### 3.7. Dietary Habits

In relation to dietary habits, Kopf et al. conducted an intervention study in humans with BMI > 25 kg/m^2^ and low intake of whole grains, fruits, and vegetables (Table 1) [65]. During the weekly interview, the subjects themselves selected the vegetables and fruits to be eaten the following week from apples, bananas, blueberries, clementines, grapes, pears, strawberries, broccoli, carrots, cauliflower, celery, green beans, green leaf lettuce, peas, spinach, sweet pepper, and tomatoes. The subjects ate these fruits and vegetables for 21 to 30 servings/week (at least three servings/day) for six weeks. As a result, compared to control group in that dietary habits were not or minimal changed, average daily intake of refined grains was 1/3, fruit intake was doubled, and vegetable intake was four times, leading to a significant reduction in blood LBP levels and IL-6 levels. An epidemiological study by Ahola et al. in patients with type 1 diabetes has shown a negative correlation between several dietary patterns and blood LPS levels: These dietary patterns are “Fish” (frequently eats fish dishes), “Healthy snack” (frequently eats fruits, berries, fresh vegetables, yoghurt, low-fat cheese, and does not drink many soft drinks) and “Modern” (frequently eats poultry, pasta, rice, meat dishes, fried and grilled foods, and fresh vegetables) (Table 2) [105]. In the epidemiological study by Ahola et al., no significant correlation was found between blood LPS levels and intake of energy, carbohydrates, fats, proteins, or dietary fiber. In regard to the absence of a significant positive correlation between blood LPS levels and fat intake (the believed cause of blood LPS elevation in humans and animals), the authors consider that the previously reported amount or proportion of fat intake may be greater than the intake in the normal diet. Similarly, Amar et al. reported no significant correlation between fat intake and blood LPS levels in 201 subjects [106]. In the same study, Amar et al. reported a positive correlation between total energy intake and blood LPS levels [106]. The effect of caloric restriction on blood LPS levels have been reported in both humans and mice. Ott et al. reported that, in women with a BMI of 30 kg/m^2^ or more, intake of a defined formula diet of 800 kcal/day for four weeks decreased blood LBP levels, and following intake of the normal diet (1800 kcal/day), blood LBP levels returned to the initial levels (Table 1) [107]. Even in mice, caloric restriction of 30% [108] or 40% [109] has been reported to decrease blood LPS or LBP levels (Table 3). A common finding in these reports in mice is that blood LPS or LBP levels are reduced by calorie restriction compared to ad libitum even in normal chow-fed mice. This suggests that the influx of LPS into the bloodstream is not limited to the specific conditions of excessive fat intake but can also occur by some mechanism in the normal diet.

## 4. Association of Dietary Factor-Induced Reduction of Blood LPS and Modulation of Gut Microbiota

Although few studies have evaluated the relationship between the effect of dietary factors on blood LPS and intestinal flora in humans, several studies have evaluated intestinal flora in oligosaccharide intervention studies (Table 1). A common finding in these reports is an increase in *Bifidobacterium*. *Bifidobacterium* has been reported to enhance the intestinal tight junction by preserving claudin 4 and occludin localization at tight junctions, and inhibit permeability in mice with colitis [115]. Similarly, in human colonic epithelial cell line T84, the addition of culture supernatant of *Bifidobacterium* has been reported to enhance barrier function through increased expression of tight junction protein, suggesting that some humoral factors contribute to improved intestinal barrier function [116]. Increased expression of tight junction protein in *Bifidobacterium*-treated mice has been reported to be associated with increased short-chain fatty acids (acetic acid, butyric acid, and propionic acid) in the intestinal tract [117]. These short-chain fatty acids have been reported in the human colonic epithelial cell line caco-2 to act as an energy source for epithelial cells to protect themselves, and also act as a histone deacetylase inhibitor which inhibit Nod-like receptor P3 inflammasomes to maintain the barrier function of epithelial cells [118]. These results suggest that the increase in *Bifidobacterium* induced by oligosaccharide intake decreases blood LPS levels through the improvement of the barrier function of the intestinal tract. In addition, dietary factors that increase *Bifidobacterium* are expected to reduce blood LPS levels.

Changes in intestinal flora by the dietary factors listed in Table 4 was greatly dependent on the study. However, all of the dietary factors commonly lowered blood LPS or LBP levels in animals, as described in Table 3. In other words, by finding bacteria that have decreased or increased in many dietary factor intervention studies, we can find specific bacteria that contribute to the increase or decrease in blood LPS levels. To this end, we have organized the number of reports that show increases or decreases of each bacterial genus (Figure 1). We selected eight of these genera (*Lactobacillus*, *Bacteroides*, *Akkermansia*, *Clostridium*, *Escherichia*, *Roseburia*, *Prevotella* and *Desulfovibrio*) as bacteria included in a sufficient number (five or more) of reports, and a biased number of reports (*Bifidobacterium* was excluded because it was discussed above. *Faecalibacterium* was also excluded because there is almost no bias in the number of reports).

*Lactobacillus*, *Bacteroides*, *Akkermansia*, *Roseburia*, and *Prevotella* are possible bacterial genera that may contribute to the reduction of blood LPS levels. *Lactobacillus* is a gram-positive bacterium that produces large amounts of lactic acid during carbohydrate fermentation. The probiotic contribution of *Lactobacillus* to the regulation of metabolic endotoxemia is studied (Table 3). Administration of *Lactobacillus rhamnosus* CNCM I-4036 to obese Zucker-Lepr*^fa^*/*^fa^* rats decreased the mRNA expression levels of endothelin receptor type B (*Ednrb*) in the intestinal mucosa, and reduced the blood LBP level [52]. Reduction of *Ednrb* decreases the density of negative charge of the colonic mucin layer, leading to an increase in the ability of the mucin layer to adsorb microparticles and bacteria, thereby inhibiting their penetration through the colonic mucosa [119]. *Lactobacillus sakei* OK67 and PK16 are reported to suppress high-fat diet-induced colitis, and to reduce the fecal *Proteobacteria* population and fecal LPS levels in mice [53]. In addition to the previous reports described in Table 3, it has been reported that oral administration of *Lactobacillus reuteri* ZJ617 suppresses LPS-induced apoptosis of intestinal epithelial cells and maintains the intestinal barrier function [120]. We have described in Section 2.2 that LPS is absorbed from the intestinal tract during lipid absorption. Interestingly, oral ingestion of *Lactobacillus acidophilus* ATCC 4356 in mice has been reported to reduce the mRNA levels of Niemann-Pick C1-like 1, which is involved in lipid absorption in the intestine, and in the suppression of cholesterol absorption [121]. Taken together, this suggests that *Lactobacillus* contributes to a decrease in blood LPS levels through strengthening the intestinal barrier, reducing the amount of LPS in feces, and suppressing lipid absorption. As described in Table 4, *Bifidobacterium*, oligofructose, galacto-oligosaccharide, syringaresinol, acaudina molpadioides-derived fucosylated chondroitin sulfate, green tea extract, Tartary buckwheat protein, nopal, semen hoveniae extract, and 30% caloric restriction are dietary factors that increase the proportion of *Lactobacillus* in the gut microbiota. Among them, the amylolytic *Bifidobacterium* strain is reported to stimulate the growth of a nonamylolytic *Lactobacillus* probably by producing intermediate metabolites of starch metabolism [122]. Oligosaccharides (oligofructose and galacto-oligosaccharide) were reported to support the growth of *Lactobacillus* as prebiotics [123]. Green tea extract [124] and buckwheat-resistant starch [125] were reported to promote the growth of *Lactobacillus* in a fermentation assay. On the other hand, in an *in vitro* fermentation assay using gut microbiota, it was reported that fucosylated chondroitin sulfate promotes the growth of *Bacteroides*, *Bifidobacterium*, and *Clostridium*, while the number of *Lactobacillus* decreases [126]. Thus, the mechanism by which *Lactobacillus* increased in mice fed with fucosylated chondroitin sulfate needs to be further studied. The mechanism by which the proportion of *Lactobacillus* in gut microbiota increases due to calorie restriction also remains unknown. As it has been reported that the bacteria adapted to the nutritional environment can grow predominantly in the gut microbiota consortium [127], *Lactobacillus* might be able to grow even under malnutrition. The effect of syringaresinol, nolpal, and semen hoveniae on the growth of *Lactobacillus* has not been revealed.

*Bacteroides* is a gram-negative obligate anaerobe. Hooper et al. reported that *Bacteroides thetaiotaomicron*, a prominent component of the normal mouse and human intestinal microflora, modulates expression of genes involved in mucosal barrier fortification [128]. The administration of *Bacteroides fragilis* HCK-B3 and *Bacteroides ovatus* ELH-B2 to mice attenuated LPS-induced intestinal inflammation, by either modulating cytokine production or restoring the Treg/Th-17 balance [129]. On the other hand, in a state in which no dietary fiber is ingested, it has been suggested that *Bacteroides* degrades the mucin layer of the intestinal tract, decreases the barrier function of mucus, and induces inflammation [130]. Therefore, it should be noted that depending on the diet of the host, *Bacteroides* can act as either probiotics or pathobionts. As described in Table 4, an increase in *Bacteroides* was reported in four out of five intervention studies with sulfated polysaccharides. *Bacteroides* is a unique bacterium among gut flora that has degrading enzymes corresponding to various sulfated polysaccharides [131] and is able to utilize sulfated polysaccharides such as heparin [131], heparan sulfate [131], and chondroitin sulfate [132] as energy sources. It is therefore thought that intake of sulfate polysaccharide preferentially nourishes *Bacteroides* in gut flora and suppresses metabolic endotoxemia via its anti-inflammatory and barrier function-enhancing effects.

*Akkermansia* is a mucin-adherent intestinal bacterium [133], which grows by degrading mucin [134], and produces propionic acid, a short-chain fatty acid [135]. In addition, *Akkermansia* promotes butyrate production, by supporting the growth of *Anaerostipes caccae* through mucin degradation [136]. As noted above, these short-chain fatty acids are known to enhance intestinal barrier function. In addition, it has been reported that *Akkermansia*-derived extracellular vesicles administered in mice are localized to the large intestine, and directly enhance intestinal barrier function by increasing epithelial cell expression of tight junction proteins [137]. Furthermore, oral administration of *Akkermansia* to mice inhibited high-fat diet-induced thinning of the mucin layer, reduced blood LPS concentration, and inhibited obesity and abnormal glucose metabolism [138]. *Akkermansia* has been reported to be negatively correlated with obesity (waist-to-hip ratio and subcutaneous adipocyte diameter) and diabetes mellitus (glucose intolerance states), and is attracting attention as a next-generation probiotic [139]. Among the dietary factors that increase the proportion of *Akkermansia* in the gut flora, polyphenols are intriguing because most of intervention studies with polyphenols (apple-derived polymeric procyanidins, genistein, and isoflavone) or polyphenol-rich food extracts (camu camu extract, cranberry extract, and green tea extract) consistently reported an increase of *Akkermansia* (Table 4). Anhê et al. reported that cranberry extract administration to mice increased colonic Kruppel-like factor 4 (a marker of goblet cells) and Muc2 mRNA expression, suggesting that polyphenols enhance mucin production and support the growth of *Akkermansia* [95]. On the other hand, direct prebiotic action of polyphenols to *Akkermansia* has been reported in a study using the Simulator of Human Intestinal Microbial Ecosystem (SHIME^®^) [140].

*Roseburia* [141] is an enteric bacterium that utilizes dietary fiber and may enhance intestinal barriers by producing butyric acid. It has been reported that administration of *Roseburia* to mice enhanced differentiation of regulatory T cells in the intestinal lamina propria and suppressed intestinal inflammation [142]. As described in Table 4, oligofructose, apple-derived polymeric procyanidins, sea cucumber-derived sulfated polysaccharide, camu camu extract, and ganoderma lucidum mycelium water extract were reported to increase the proportion of *Roseburia* in the gut flora. *Roseburia* metabolizes oligofructose into fructose, which is used for growth, but for this process, acetic acid that is produced by *Bifidobacterium* is required [143]. Therefore, in order to grow *Roseburia* by oligofructose intake, it is necessary to pay attention to the symbiotic relationship with other intestinal bacteria and the amount of short-chain fatty acids in the intestine. Other dietary factors, procyanidins, sea cucumber-derived sulfated polysaccharide, camu camu extract, and ganoderma lucidum mycelium, have not been studied for their prebiotic function for *Roseburia*.

It has been suggested that LPS from *Prevotella* has fewer phosphate and acyl moieties contributing to endotoxin activity, resulting in a lower TLR4 stimulatory capacity than LPS from *Salmonella* [144]. Therefore, by increasing the population of *Prevotella* in the intestinal flora, endotoxin activity in the intestinal contents and damage to intestinal epithelial cells might be decreased, leading to the reduction of blood LPS levels. On the other hand, *Prevotella* produces succinate as a metabolite of sugar metabolism [145]. It has also been reported that succinate from intestinal bacteria is utilized by and promotes growth of *Salmonella* serovar Typhimurium [146] and *Clostridium difficile* [147], which are the pathogens of pseudomembranous colitis. Succinate has also been reported to induce colitis via succinate receptors and to promote colonic fibrosis [148]. In addition, proportion of *Prevotella* in the gut flora has been reported to be positively correlated with blood LPS levels in patients with type 2 diabetes [149]. Thus, an increase in the proportion of *Prevotella* does not necessarily have a positive effect on intestinal health. It is necessary to carefully investigate the contribution of *Prevotella* to blood LPS levels.

*Clostridium*, *Escherichia*, and *Desulfovibrio* are bacterial genera that may contribute to the increase of blood LPS levels. Many pathogenic bacteria (such as enterohemorrhagic *Escherichia coli*, *Clostridium botulinum*, *Clostridium tetani*, and *Clostridium perfringens*), which produce effector proteins or enterotoxins that disrupt epithelial tight junction belong to these genera [150]. In addition, the endotoxin activity of LPS in non-pathogenic *Escherichia* is also higher than in *Bacteroides*, and an increased proportion of these *Escherichia* in enteric flora aggravate colitis [151]. *Clostridium* species catabolize cholic acid to deoxycholic acid for their growth [152]. It is reported that, in mice, deoxycholic acid increases intestinal permeability through the reduction of goblet cell number, suppression of mucin production, induction of low-grade inflammation, and suppression of tight junction protein (ZO-1) expression [153]. In terms of dietary factors that reduce *Escherichia*, there are many reports of sulfated polysaccharides (Table 4). We could not find any reports that suggested the direct inhibitory effect of sulfated polysaccharide on growth of *Escherichia*. On the other hand, it is suggested that *Bacteroides*, that can be preferentially grown in sulfated polysaccharide feeding, compete with *Escherichia* in the co-culture assay [127]. In order to elucidate the mechanism by which sulfated polysaccharides reduce the proportion of *Escherichia*, it is hoped to study focusing on the interaction between gut microbes. Among the dietary factors that reduce *Clostridium*, procyanidin is reported to decrease the growth of *Clostridium* in fecal batch culture [154]. The bactericidal activity of methanol extract of nopal against Clostridium has also been reported [155].

*Desulfovibrio* is a gram-negative, obligate anaerobe, sulfate-reducing bacterium. *Desulfovibrio* utilizes electrons supplied by the oxidation of lactic acid in the electron transport system of the respiratory chain, uses sulfuric acid as the final electron acceptor, and produces hydrogen sulfide as a metabolite [156]. *Desulfovibrio* is ubiquitous in the intestines of humans and mice. Of the studies that showed significant changes in the proportion of *Desulfovibrio*, most studies reported that the proportion was increased associated to the reduction of blood LPS levels (Table 4). However, it is also reported that proportions of *Desulfovibrio* increased in the colons of patients with ulcerative colitis [157] and has attracted attention as a pathogen of colitis. In addition, Xie et al. reported in mice that the increase of *Desulfovibrio* in feces was positively correlated with the increase of LPS levels in feces, liver, and blood [45]. Qui et al. reported that ingestion of a high-fat diet in mice increased fecal *Clostridium* and *Desulfovibrio*, and oral administration of these bacteria to the normal chow-fed mice increased fecal and blood LPS levels [158]. These reports suggest that *Desulfovibrio* plays an important role as a source of LPS in the intestine. *Desulfovibrio* also competes with *Anaerostipes caccae* for lactic acid produced by *Bifidobacterium*, and reduces butyric acid production by inhibiting the growth of *Anaerostipes caccae* [159]. In addition, as the coexistence of *Desulfovibrio* and *Bifidobacterium* inhibits the growth of *Bifidobacterium* [159], this suggests that the amount of acetic acid produced by *Bifidobacterium* might be also reduced. On the other hand, it has also been reported that oral administration of *Desulfovibrio* increases the amount of hydrogen sulfide in the intestinal tract and inhibits intestinal peristalsis [160]. *Desulfovibrio* is thought to play an important limiting role in increasing blood LPS levels by supplying LPS, decreasing intestinal barrier function due to reduction of short-chain fatty acid content, and prolonging retention time of intestinal contents due to inhibition of peristalsis (Figure 2). However, despite *Desulfovibrio* being an important target for metabolic endotoxemia, few dietary factors have been reported to reduce the proportion of *Desulfovibrio* (Table 1, Table 2, Table 3 and Table 4).

Sulforaphane (1-isothiocyanato-4-methylsulfinylbutane) is an isothiocyanate with an N=C=S functional group and is abundant in broccoli (especially the sprout) and other cruciferous vegetables as the precursor glucoraphanin. Sulforaphane is thought to play a role in plant protection through its antimicrobial action [161], induction of programmed cell death of infected tissue [162], and inhibition of insect feeding [163]. On the other hand, in humans and rodents, sulforaphane activates NF-E2-related factor 2 (NRF2), which induces expression of genes expressing antioxidant and detoxication enzymes, including phase II enzymes, and then exerts anti-cancer [164], anti-liver damage [165], and anti-depressive effects [166].

We found that dietary administration of broccoli sprout extract reduced blood LPS levels and attenuated obesity, glucose intolerance, hepatic steatosis, and inflammation in mice fed a high-fat diet [94]. We also reported that the proportion of *Desulfovibrionaceae* [upper taxa (family) of *Desulfobivrio*] was positively correlated with blood LPS levels, and that ingestion of broccoli sprout extract reduced *Desulfovibrionaceae* in cecal contents. Subsequently, Wu et al. also reported that broccoli powder reduced the proportion of *Desulfovibrio* in the large intestinal contents of mice [167]. They reported that the decrease in *Desulfovibrio* composition was negatively correlated with the activity of myrosinase-like activity, isothiocyanate content, and NAD(P)H:quinone dehydrogenase 1 (NQO1) in the colonic mucosa. Ingested glucoraphanin is metabolized by myrosinase-like enzymes in enteric bacteria, which then produce sulforaphane [168]. Since sulforaphane enhances NQO1 activity through activation of NRF2 [168], it is suggested that sulforaphane metabolized and formed from glucoraphanin in broccoli sprouts may have an inhibitory effect on *Desulfovibrio* (Figure 2). Sulforaphane has been reported to exert antibacterial activity against the *Proteobacteria* (*Desulfovibrio* belongs this phyla) [169], but its direct effect on *Desulfovibrio* is not well understood. It is hoped that the mechanism by which sulforaphane decreases the proportion of *Desulfovibrio* will be elucidated.

## 5. Conclusions

In this article, we summarized previous reports about the regulation of metabolic endotoxemia through dietary factors, focusing on gut microbiota. Although changes in the composition of *Firmicutes* and *Bacteroides* due to excessive fat intake have been reported to contribute to metabolic endotoxemia in many reports, the results differ between studies and between species, and further investigation is needed to find true pathobionts. Moreover, since human epidemiological studies have not found a correlation between fat intake and blood LPS levels, it is necessary to search for dietary factors other than fat that cause metabolic endotoxemia. Regarding dietary factors that improve metabolic endotoxemia, human intervention studies have focused on probiotics, prebiotics, polyphenols and dietary habits, and it has been reported that prebiotics, including oligosaccharides, are effective. On the other hand, few studies have evaluated the effects of dietary intervention on gut flora in humans. The development and popularization of next-generation sequencing has made it possible to comprehensively analyze the “fecal” microbiota in humans. On the other hand, as mentioned above, there are also mucin-adherent bacteria that are thought to be involved in metabolic endotoxemia (e.g., *Akkermansia* and *Bacteroides*). In a colitis mouse model, it has been reported that the bacterial flora in the mucin layer exhibits changes from 12 weeks before the onset of colitis, and that the mucin layer was thinned [170]. In this study, changes in the fecal flora occurred at the same time as the onset of colitis, indicating that the bacteria in the mucin layer play an important role in understanding the physiological state of the intestinal tract. However, although it is possible to collect mucin layer samples in animals, it is not easy to do so in humans, due to ethical and technical obstacles. In the future, if a method for collecting the mucin layer in a noninvasive manner is established in humans, the research field of metabolic endotoxemia can be further advanced. Then, it is expected that we will comprehensively understand the relationship between dietary factors, dysbiosis, and metabolic endotoxemia in humans by conducting human intervention studies and epidemiological studies with dietary surveys, gut microbiota analysis using next-generation sequencers and evaluation of blood LPS levels.

## Figures and Tables

**Figure 1 nutrients-11-02277-f001:**
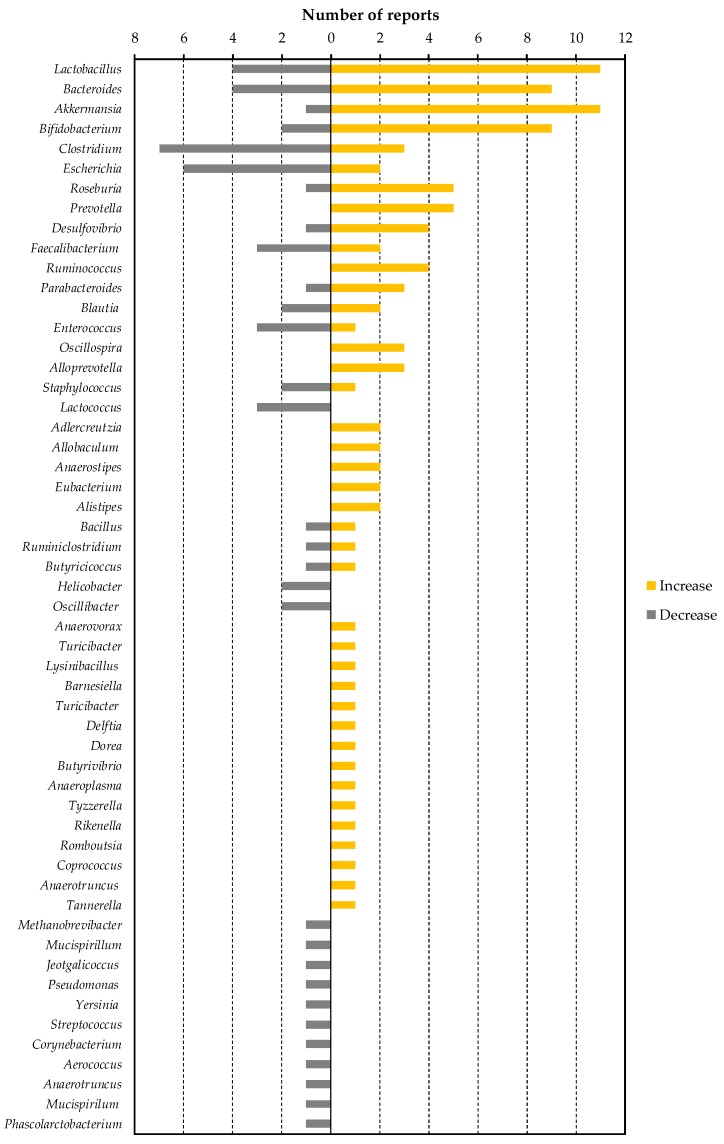
The number of reported changes of intestinal bacterial genera in dietary factor intervention studies in animals.

**Figure 2 nutrients-11-02277-f002:**
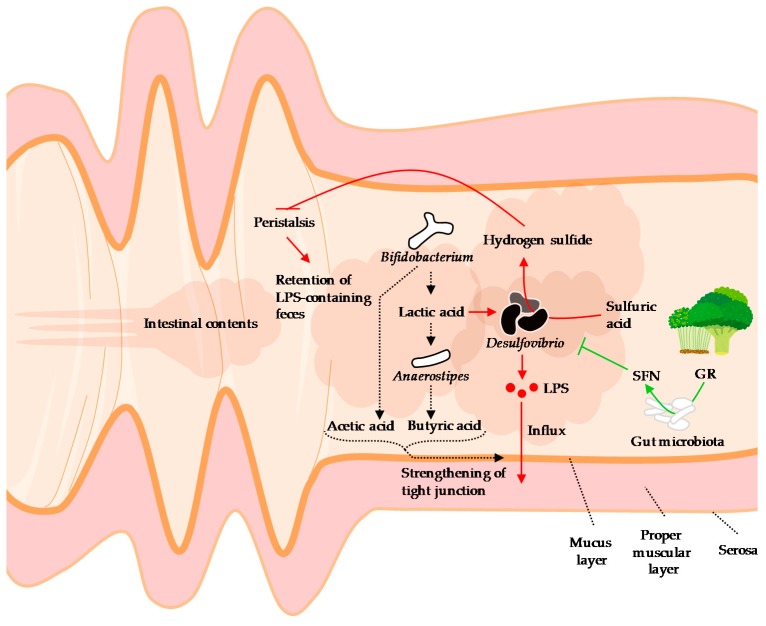
A hypothetical schematic of the behavior of *Desulfovibrio* in the intestine, influx of LPS, and the effects of sulforaphane on *Desulfovibrio*. *Desulfovibrio*, a source of LPS, reduces the amount of short-chain fatty acids in the intestinal tract through lactic acid consumption and suppression of growth of *Bifidobacterium*, thereby attenuating tight junction. In addition, hydrogen sulfide, a metabolite of *Desulfovibrio*, inhibits peristalsis, thereby retaining the LPS-containing intestinal contents and promoting LPS absorption. The functional component of the broccoli sprouts, glucoraphanin, is metabolized by enteric bacteria to sulforaphane. Sulforaphane inhibits the growth of *Desulfovibrio* and the entry of LPS into the blood (details are described in the Section 4). LPS: Lipopolysaccharide, GR: Glucoraphanin, SFN: Sulforaphane. Broccoli and sprouts illustrations © irasutoya, 2012 and 2013, respectively.

**Table 1 nutrients-11-02277-t001:** Dietary factors that have been evaluated for efficacy on blood lipopolysaccharide (LPS) levels in human interventional studies.

Category	Dietary Factor	Dose	Consumption Period	Subject	LPS	LBP	Gut Microbes with Significant Changes in Proportion **	
							Increase	Decrease
Probiotics/ Prebiotics	Yakult light (*Lactobacillus casei* Shirota 1 × 10^8^ CFU/mL) [48]	195 mL	3 months	Metabolic syndrome	ND	↑	—	—
	Low-fat yogurt [49]	339 g	9 weeks	Healthy subject or Obesity	→	→	—	—
	Low-fat yogurt [50]	226 g	Premeal	Healthy subject or Obesity (postprandial endotoxemia was assessed)	→	→	—	—
	Oligofructose [57]	21 g	12 weeks	Overweight/ Obesity	↓	—	—	—
	Oligofructose- enriched inulin [58]	10 g	8 weeks	Type 2 diabetes	↓	—	—	—
	Inulin + Oligofructose [59]	8 g 8 g	3 months	Obesity	→	—	*Bifidobacterium*, *Faecalibacterium prausnitzii*	*Bacteroides intestinalis*, *Bacteroides vulgatus*, *Propionibacterium*
	Galacto- oligosaccharide [60]	5.5 g	12 weeks	Type 2 diabetes	→	→	none	none
	Galacto- oligosaccharide [61]	15 g	12 weeks	Overweight/ Obesity	—	→	*Bifidobacterium* spp.	none
	α-Galacto- oligosaccharide [62]	6–18 g	14 days	Overweight	↓	—	Bifidobacteria	none
	Resistant dextrin [63]	10 g	8 weeks	Type 2 diabetes	↓	—	—	—
	Insoluble dietary fiber [from Fiber One Original cereal (General mills)] [64]	30 g	With high-fat, high-calorie meal	Healthy subject (postprandial endotoxemia was assessed)	↓*	—	—	—
	Whole grains [65]	3 servings	6 weeks	Overweight/ Obesity	—	↓	none	none
Probiotics/ Prebiotics	*Bifidobacterium longum* + Oligofructose + Life style modification [110]	—	24 weeks	Non-alcoholic steatohepatitis	↓	—	—	—
Polyphenol	Resveratrol + Polyphenol [72]	100 mg 75 mg	10 minutes before intake of high-fat high-carbohydrate meal	Healthy subjects (postprandial endotoxemia was assessed)	—	↓ *	—	—
	Red wine [73]	272 mL	With high-fat meal	Healthy subjects (postprandial endotoxemia was assessed)	→	→	—	—
Dietary habits	Fruits + Vegetables [65]	3 servings	6 weeks	Overweight/ Obesity	—	↓	α-diversity (No significant change in bacterial genera was found)	none
	Caloric restriction [107]	800 kcal	4 weeks	Obesity	—	↓	*Anaerostipes hadrus*, *Blautia* sp., *Ruminococcus faecis*, *Bifidodbacterium* sp.	*Agathobacter rectalis*
Others	Glutamine [85]	30 g	14 weeks	Overweight/ Obesity	↓	—	—	—

ND: Not detected, —: No data, ↑: Significantly increased, →: Not significantly changed, ↓: Significantly decreased, *: Attenuation of postprandial endotoxemia, **: The bacteria mentioned by the author in the paper are listed.

**Table 2 nutrients-11-02277-t002:** Correlation of dietary factors, gut microbes, and blood LPS levels in human epidemiological studies.

Subject	Number of Subject	Correlation of Dietary Factor and Gut Microbe *			Correlation of Blood LPS and Gut Microbe			Correlation of Blood LPS and Dietary Factor		
Over- weight pregnant women [111]	88	*P*	Dietary fiber vs.	diversity, richness, *Firmicutes* in unidentified family of order *Clostridiales*, *Barnciellaceae* family belonging to the phylum *Bacteroidetes*	*P*	none		*P*	none	
			Vitamin A, β-Carotene vs.	*Firmicutes*						
		*N*	Fat vs.	diversity, richness, *Barnsiellaceae*	*N*	none		*N*	none	
Healthy subjects [86]	150	*N*	25-Hydroxy vitamin D vs.	*Coprococcus*, *Bifdobacterium*	*N*	LPS vs.	*Faecalibacterium*	*N*	LPS vs.	25-Hydroxy vitamin D
Type 1 diabetes [105]	668	—	—	—	—	—	—	*N*	LPS vs.	Dietary pattern; “Fish”(frequently eat fish dishes), “Healthy snack” (frequently eat fruits, berries, fresh vegetable, yoghurt, low-fat cheese, and do not drink much soft drinks), “Modern”(frequently eat poultry, pasta, rice, meat dishes, fried and grilled foods, and fresh vegetables)

—: No data, *P*: Positive correlation, *N*: Negative correlation, LPS: lipopolysaccharide, *: The bacteria mentioned by the author in the paper are listed.

**Table 3 nutrients-11-02277-t003:** Dietary factors that have been evaluated for efficacy on blood LPS levels in animal interventional studies.

Category	Dietary Factor	Dose	Administration Period	Model	LPS	LBP	Significant Change in Gut Microbiota
Probiotics/ Prebiotics	*Lactobacillus rhamnosus* GG [51]	1 × 10^8^ CFU/day	12 weeks	HFD-fed ApoE KO mouse	↓	—	no
	*Lactobacillus rhamnosus* CNCM I-4036 [52]	1 × 10^10^ CFU/day	30 days	Chow diet-fed Zucker-Lep*^fa^*^/^*^fa^* rat	—	↓	—
	*Lactobacillus sakei* OK67 +/− *Lactobacillus sakei* PK16 [53]	1 × 10^9^ CFU/day 1 × 10^9^ CFU/day	4 weeks	HFD-fed C57BL/6 mouse	↓	—	yes
	*Bifidobacterium longum* BR-108 (sterilized) [54]	200, 400 mg/kg/day	4 weeks	HFD-fed C57BL/6J mouse	↓	—	yes
	*Bifidobacterium infantis* + *Lactobacillus acidophilus* + *Bacillus cereus* [55]	0.5 × 10^6^ CFU/day 0.5 × 10^6^ CFU/day 0.5 × 10^5^ CFU/day	12 weeks	HFHSD-fed SD rat	↓	—	yes
	*Lactobacillus plantarum* LC27 +/− *Bifidobacterium longum* LC67 [43]	1 × 10^9^ CFU/day each (or 0.75 × 10^9^ (LC27) + 0.25×10^9^ (LC67) CFU/day in mix)	4 weeks	HFD-fed C57BL/6 mouse	↓	—	yes
	Oligofructose [112]	10% (mixed in diet)	12 weeks	HFHSD-fed SD rat	↓	—	yes
	Galacto- oligosaccharide [84]	800 mg/kg/day	8 weeks	HFD-fed SD rat	↓	—	yes
	Inulin [113]	5% (intragastric administration, sample volume was not described)	6 weeks	standardized diet (kcal %: 10% fat, 20% protein, and 70% carbo- hydrate; 3.85 kcal g^−1^)-fed *db*/*db* mouse	↓	—	yes
	Wheat-derived arabinoxylan [114]	7.5% (mixed in diet)	8 weeks	HFD-fed C57BL/6J mouse	↓	—	—
Polyphenols	Grape seed proanthocyanidin [33]	500 mg/kg/day	10 days (prophylactic) or 17 weeks (with cafeteria diet)	Cafeteria diet (high-fat/high carbohydrate diet)-fed Wistar rat	↓	—	—
	Grape-seed proanthocyanidin [29]	100, 500 mg/kg/day	2 weeks	Cafeteria diet (high saturated-fat/high refined-carbohydrate diet)-fed Wistar rat	↓	—	—
	Resveratrol [74]	50, 75, 100 mg/kg/day	16 weeks	HFD-fed C57BL/6 mouse	↓	↓	yes
	Apple-derived polymeric procyanidins [75]	0.5% (administration route was not described)	20 weeks	HFHSD-fed C57BL/6J mouse	↓	—	yes
	Genistein [76]	0.2% (mixed in diet)	6 months	HFD-fed C57BL/6 mouse	↓	—	yes
	Isoflavone [77]	0.1% (mixed in diet)	5 weeks	HFD-fed C57BL/6 mouse	↓	↓	yes
	Syringaresinol [78]	50 mg/kg/day	10 weeks	40-week-old C57BL/6 mouse	—	↓	yes
Sulfated polysaccharide	Sea cucumber-derived sulfated polysaccharide [80]	300 mg/kg/day	8 weeks	HFD-fed BALB/c mouse	—	↓	yes
	Sea cucumber-derived sulfated polysaccharide [81]	300 mg/kg/day	42 days	Chow-fed BALB/c mouse	—	↓	yes
	Acaudina molpadioides-derived fucosylated chondroitin sulfate [82]	80 mg/kg/day	10 weeks	HFD-fed C57BL/6J mouse	↓	—	yes
	Chicken-derived chondroitin sulfate [83]	150 mg/kg/day	16 days	Exhaustive exercise stress model BALB/c mouse	↓	—	yes
	Fucoidan [84]	100 mg/kg/day	8 weeks	HFD-fed SD rat	↓	—	yes
Other dietary components	Tetrahydro iso-alpha acid (included in hops) [88]	0.1% (mixed in diet)	8 weeks	HFD-fed C57BL/6J mouse	↓	—	—
	Rhein (included in rhubarb) [89]	120 mg/kg/day	6 weeks	HFD-fed C57BL/6J mouse	↓	—	yes
	Phlorizin (included in apple) [90]	20 mg/kg/day	10 weeks	Chow-fed *db*/*db* mouse	↓	—	yes
	Capsaicin [91]	0.01% (mixed in diet)	12 weeks	HFD-fed C57BL/6J mouse	↓	—	yes
	Rutin [92]	0.64% (mixed in diet)	20 weeks	HFD-fed C57BL/6J mouse	↓	—	yes
	Lycopene [93]	0.03% (mixed in diet)	10 weeks	HFD and fructose-fed C57BL/6 J mouse	↓	—	—
Other extracts/dietary components	Broccoli sprout extract [94]	2.2% (mixed in diet)	14 weeks	HFD-fed C57BL/6JSlc mouse	↓	↓	yes
	Camu camu extract [96]	200 mg/kg/day	8 weeks	HFHSD-fed C57BL/6J mouse	↓	—	yes
Other extracts/dietary components	Cranberry extract [95]	200 mg/kg/day	8 weeks	HFHSD-fed C57BL/6J mouse	↓	—	yes
	Green tea extract [34]	2% (mixed in diet)	8 weeks	HFD-fed C57BL/6J mouse	↓	—	yes
	Tartary buckwheat protein [32]	23.5% (mixed in diet)	6 weeks	HFD-fed C57BL/6 mouse	↓	—	yes
Foods	Cocoa [97]	8% (mixed in diet)	18 weeks	HFD-fed C57BL/6J mouse	↓	—	—
	Nopal [98]	5% of dietary fiber was replaced with those of nopal-derived (mixed in diet)	1 month	HFHSD-fed Wistar rat	↓	—	yes
	Steamed fish meat [99]	*Ad libitum* (9:00–12:00 and 18:00–21:00)	8 weeks	Chow-fed C57BL/6 mouse	—	↓	yes
Chinese medicines	Geniposide + Chlorogenic acid [100]	90 mg/kg/day 1.34 mg/kg/day	4 weeks	HFD-fed C57BL/6 mouse	—	↓	—
	Potentilla discolor Bunge water extract [101]	400 mg/kg/day	8 weeks	HFD-fed, streptozotocin- injected C57BL/6J mouse	↓	↓	yes
	Ganoderma lucidum mycelium water extract [102]	2–8 mg/day	8 weeks	HFD-fed C57BL/6NCrlBltw mouse	↓	—	yes
	Semen hoveniae extract [103]	300, 600 mg/kg/day	8 weeks	Alcohol-containing Lieber-DeCarli diet-fed SD rat (Alcoholic liver disorder model)	↓	—	yes
	Shenling Baizhu powder [104]	30 g/kg/day	16 weeks	HFD-fed SD rat	↓	—	yes
Caloric restriction	30% caloric restriction [108]	—	62–141 weeks	HFD, LFD-fed C57BL/6J mouse	—	↓	yes
	40% caloric restriction [109]	—	30 days	Chow-fed C57BL/6J mouse	↓	↓	yes

—: No data, HFD: High-fat diet, HFHSD: High-fat high-sucrose diet, ↑: Significantly increased, →: Not significantly changed, ↓: Significantly decreased.

**Table 4 nutrients-11-02277-t004:** Changes of gut microbiota induced by dietary factor intervention in animal experiments.

Category	Dietary Factor	Sample	Method	Gut Microbe with Significant Changes in Proportion *	
				Increase	Decrease
Probiotics/ Prebiotics	*Lactobacillus sakei* OK67 +/− *Lactobacillus sakei* PK16 [53]	Feces	PCR, NGS	OTU (O67), Ace (O67), Chao1 (O67), Shanon (O67)	Simpson (O67), *Proteobacteria*, *Firmicutes*, *Firmicutes*/*Bacteroidetes*, *Proteobacteria*/*Bacteroidetes*
	*Bifidobacterium longum* BR-108 (sterilized) [54]	Cecal contents	PCR	*Bifidobacterium* spp., *Lactobacillus* spp.	*Firmicutes*
	*Bifidobacterium infantis* + *Lactobacillus acidophilus* + *Bacillus cereus* [55]	Feces	PCR	Bifidobacteria, *Lactobacillus*, *Bacteroides*, Bifidobacteria/*Escherichia coli*	*Escherichia coli*, *Enterococcus*
	*Lactobacillus plantarum* LC27 +/− *Bifidobacterium longum* LC67 [43]	Feces	PCR	*Actinobacteria* (LC67, LC27 + LC67)	*Firmicutes*, *Bacteroidetes*, δ/γ-*Proteobacteria*, *Deferribacteres* (LC67, LC27+LC67), *Firmicutes*/*Bacteroidetes*, *Proteobacteria*/*Bacteroidetes*
	Oligofructose [112]	Cecal contents	PCR	*Bacteroides*/*Prevotella*, *Bifidobacterium*, *Lactobacillus*, *Roseburia*	*Clostridium leptum* (cluster IV), *Clostridium* cluster I, *Clostridium* cluster XI, *Methanobrevibacter*, *Akkemansia muciniphila*, *Faecalibacterium prausnitzii*
	Galacto-oligosaccharide [84]	Cecal contents	NGS	*Verrucomicrobia*, *Akkermansia*, *Ruminococcus*, *Blautia*, *Bacteroidetes*, *Proteobacteria*, *Adlercreutzia*, *Staphylococcus*, *Prevotella*, *Oscillospira*, *Lactobacillus*, *Desulfovibrio*	*Firmicutes*, *Actinobacteria*, *Clostridium*, *Bacillus*
	Inulin [113]	Feces	NGS	*Bacteroidetes*, *Cyanobacteria*, *Bacteroides*	*Firmicutes*, *Deferribacteres*, *Tenericutes*, *Ruminiclostridium*_6, *Mucispirillum*
Polyphenols	Resveratrol [74]	Cecal contents	NGS	*Deferribacteraceae*	none (In this study, population of *Desulfovibrionaceae* in the high-fat diet + intervention group was at the same level with normal chow group, but there was no significant reduction from high-fat diet group.)
	Apple-derived polymeric procyanidins [75]	Cecal contents	NGS	*Bacteroidetes*, *Verrucomicrobia*, *Adlerceitzia*, *Roseburia*, S24-7, *Bacteroids*, *Anaerovorax*, rc4-4, *Akkermansia*	*Firmicutes*, *Firmicutes*/*Bacteroidetes*, *Clostridium*, *Lachnospiraceae*, *Bifidobacterium*
Polyphenols	Genistein [76]	Feces	NGS	*Firmicutes*, *Verrucomicrobia*, *Prevotellaceae*, *Verrucomicrobia*, *Prevotella*, *Akkermansia*, *Faecalibacterium*, *Prevotella copri*, *Prevotella stercorea*, *Akkermansia muciniphila*	*Bacteroidetes*, *Bacteroidaceae*, *Bacteroides*, *Bacteroides acidifaciens*, *Bacteroides uniformis*
	Isoflavone [77]	Feces	NGS	α-diversity, *Actinobacteria*, *Verrucomicrobia*, *Bifidobacterium*/*Enterobacteriaceae*, *Akkermansia*	*Proteobacteria*
	Syringaresinol [78]	Cecal contents	NGS	*Firmicutes*/*Bacteroidetes*, *Firmicutes*, *Lactobacillus*, *Lactobacillus animalis*, *Lactobacillus johnsonii*, *Lactobacillus reuteri*, *Lactobacillus intestinalis*, *Bifidobacterium pseudolongum*	Shannon diversity indices, *Jeotgalicoccus nanhaiensis*, *Staphylococcus lentus*, *Bacteroidaceae* (EF098405_s), *Bacteroides vulgatus*, *Akkermansia muciniphila*
Sulfated polysaccharide	Sea cucumber-derived sulfated polysaccharide [80]	Feces	NGS	bacterial diversity, *Verrucomicrobia* (depolymerized sulfated polysaccharide), *Bacteroides*, *Alloprevotella*, *Ruminiclostridium*_9, *Butyricicoccus*, *Akkermansia*	*Proteobacteria*, *Escherichia*-*Shigella* (polymerized sulfated polysaccharide), *Pseudomonas* (depolymerized sulfated polysaccharide), *Yersinia* (depolymerized sulfated polysaccharide), (In this study, decrease of *Desulfovibrio* with the intervention of sulfated polysaccharide to high-fat diet-fed mouse was shown as heatmap, but significance of difference was not described.)
	Sea cucumber-derived sulfated polysaccharide [81]	Feces	NGS	*Proteobacteria* (polymerized sulfated polysaccharide), *Bacteroides* (polymerized sulfated polysaccharide), *Allobaculum* (depolymerized sulfated polysaccharide), *Alloprevotella*, *Roseburia*, *Turicibacter*, *Desulfovibrio*	*Enterococcus*, *Streptococcus*, *Escherichia*-*Shigella*, *Lactobacillus*
	Acaudina molpadioides-derived fucosylated chondroitin sulfate [82]	Feces	PCR, NGS	*Bacteroidetes*, *Lactobacillus*, *Actinobacteria*, *Faecalibacterium prausnitzii*, *Deferribacteres*, *Bacteroidales*, *Bifidobacteriales*, *Lachnospiraceae* NK4A136 group, *Bacteroides*, *Bacteroides acidifaciens*, *Bifidobacterium choerinum*	*Firmicutes*, *Escherichia coli*, *Clostridiales*, *Bacilli*, *Lactobacillales*, *Clostridia Clostridiales*, *Firmicutes Clostridiales*, *Lactococcus*, *Clostridium ruminantium*
	Chicken-derived chondroitin sulfate [83]	Feces	NGS	*Bacteroidetes*, *Bacteroides acidifaciens*, family S24-7, *Lysinibacillus boronitolerans*	*Firmicutes*, β-*Proteobacteria*
	Fucoidan [84]	Cecal contents	NGS	*Proteobacteria*, *Verrucomicrobia*, *Enterobacter*, *Bacteroidetes*, *Bacillus*, *Ruminococcus*, *Adlercreutzia*, *Prevotella*, *Oscillospira*, *Desulfovibrio*,	*Firmicutes*, *Actinobacteria*, *Clostridium*, *Corynebacterium*, *Staphylococcus*, *Lactobacillus*, *Aerococcus*
Other dietary components	Rhein (included in rhubarb) [89]	Cecal contents	PCR	*Bacteroides*/*Prevotella*, *Desulfovibrio*	*Bifidobacterium*, *Lactobacillus*
	Phlorizin (included in apple) [90]	Feces	PCR, DGGE	*Akkermansia muciniphila*, *Prevotella*	none
	Capsaicin [91]	Cecal contents	NGS	*Ruminococcaceae*, *Lachnospiraceae*	family S24_7
	Rutin [92]	Small intestinal contents	NGS	*Bacteroidales*_S24-7 group, *Bacteroidaceae*, *Porphyromonadaceae*, *Rikenellaceae*, *Desulfovibrionaceae*	*Firmicutes*, *Firmicutes*/*Bacteroidetes*, *Deferribacteraceae*, *Lachnospiraceae*
Other extracts/dietary components	Broccoli sprout extract [94]	Cecal contents	NGS	none	*Proteobacteria*, *Desulfovibrionaceae*
	Camu camu extract [96]	Feces	NGS	microbial richness, *Bifidobacterium*, *Barnesiella*, *Barnesiella* spp., *Turicibacter* spp., *Akkermansia muciniphila*, *Delftia*, *Roseburia*, *Anaerostipes*, unclassified genera within the families *Christensenellaceae*, unclassified genera within the families *Erysipelotrichaceae*	*Firmicutes*/*Bacteroidetes*, *Lactobacillus*, *Anaerotruncus*, *Parabacteroides*
	Cranberry extract [95]	Feces	PCR, NGS	*Akkermansia*	none
	Green tea extract [34]	Cecal contents	NGS	Shannon index, Chao1 richness, *Bacteroidetes*, *Actinobacteria*, *Verrucomicrobia*, *Bacteroidales*, *Bifidobacteriales*, *Verrucomicrobiales*, *Turicibacterales*. RF39, *Coriobacteriales*, *Bifidobacterium*, *Blautia*, *Dorea*, *Lactobacillus*, *Ruminococcus*, *Akkermansia*, *Butyrivibrio*, *Akkermansia muciniphila*, *Ruminococcus gnavus*, *Bifidobacterium pseudolongum*, *Bifidobacterium adolescentis*	*Firmicutes*, *Firmicutes*/*Bacteroidetes*, *Clostridiales*, SMB53
	Tartary buckwheat protein [32]	Feces	PCR	*Bifidobacterium*, *Lactobacillus*, *Enterococcus*, *Clostridium*	*Escherichia coli*, *Bacaeroides*
Foods	Nopal [98]	Feces	NGS	α-diversity, *Anaeroplasma*, *Prevotella*, *Ruminucoccus*, *Bacteroides fragilis*, *Ruminococcus bromii*, *Rumminococcus flavefaciens*, *Lactobacillus reuteri*, *Akkermansia muciniphila*	*Firmicutes*/*Bacteroidetes*, *Faecalibacterium*, *Clostridium*, *Butyricicoccus*, *Bacteroides acidifaciens*, *Blautia producta*, *Faecalibacterium prausnitzii*, *Butyricicoccus pullicaecorum*, *Clostridium citroniae*
	Steamed fish meat [99]	Feces	NGS	*Proteobacteria*, *Firmicutes*, *Ruminococcaceae*, *Oscillospira*, *Clostridium*, *Escherichia*	Shannon index, *Bacteroidetes*, S24-7
Chinese medicines	Potentilla discolor Bunge water extract [101]	Feces	NGS	*Bacteroidetes*, *Bacteroidales*_S24-7_group, norank_f_*Bacteroidales*_S24-7_group, *Parabacteroides*, *Eubacterium*_*nodatum*_group, norank_f_*Rhodospirillaceae*, *Tyzzerella*, *Rikenella*, *Alistipes*, *Lachnospiraceae*_NK4A136_group, norank_f_*Ruminococcaceae*, *Romboutsia*, *Coriobacteriaceae*_UCG_002, *Bacteroides*, *Allobaculum*, *Coprococcus*_3, norank_f_*Christensenellaceae*	*Proteobacteria*, *Helicobacteraceae*, *Helicobacter*
	Ganoderma lucidum mycelium water extract [102]	Cecal contents	NGS	*Parabacteroides goldsteinii*, *Bacteroides* spp., *Anaerotruncus colihominis*, *Roseburia hominis*, *Clostridium methylpentosum* (*Clostridium* IV), *Clostridium* XIVa, *Clostridium* XVIII, *Eubacterium coprostanoligenes*	*Firmicutes*/*Bacteroidetes*, *Proteobacteria*, *Mucispirilum shaedleri*, *Escherichia fergusonii*, *Enterococcus* spp., *Lactococcus lactis*, *Clostridium lactatifermentans* (*Clostridium* XIVb), *Oscillibacter valericigenes*
	Semen hoveniae extract [103]	Feces	NGS	Shannon index, *Verrucomicrobia*, *Bacteroidetes*, *Parabacteroides*, *Alloprevotella*, *Alistipes*, *Lactobacillus*, *Akkermansia*	*Proteobacteria*, *Firmicutes*/*Bacteroidetes*, *Oscillibacter*, *Helicobacter*
Chinese medicines	Shenling Baizhu powder [104]	Feces	NGS	Shannon index, *Actinobacteria*, *Cyanobacteria*, *Anaerostipes*, *Bifidobacterium*	*Firmicutes*/*Bacteroidetes*, *Blautia*, *Roseburia*, *Phascolarctobacterium*, *Desulfovibrio* (Significance of difference was not described)
Caloric restriction	30% caloric restriction [108]	Feces	NGS	(low-fat diet vs. low-fat diet with caloric restriction) *Lactobacillus*, OTU45 (in *Lactobacillus*), *Bifidobacterium*, [increased by caloric restriction with both of low-fat diet or high-fat diet] OTU119, OTU155, OTU267 (in *Tannerella*)	(low-fat diet vs. low-fat diet with caloric restriction) *Streptococcaceae*, TM7, OTU469 (in *Desulfovibrionaceae*) [decreased by caloric restriction with both of low-fat diet or high-fat diet] OTU65 (in *Lactococcus*), OTU366 (in *Bacteroidales*), OTU37 (in *Peptostreptococcaceae*),
	40% caloric restriction [109]	Feces	NGS	*Lactobacillaceae*, *Erysipelotichaceae*, *Bacteroidaceae*, *Verrucomicrobiaceae*	*Firmicutes*

PCR: Polymerase chain reaction, NGS: Next-generation sequencing, DGGE: Denaturing gradient gel electrophoresis, OUT: Operational taxonomic unit, *: The bacteria mentioned by the author in the paper are listed.

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
