# Peer review of "Regulation of Gut Microbiota and Metabolic Endotoxemia with Dietary Factors"

_nutrients, 2019, doi:10.3390/nu11102277_

Round 1
Reviewer 1 Report
This is a well-written review with a focus on how dietary factors (fat intake, probiotics, etc) influence the gut microbiota and metabolic endotoxemia. A few suggestions to help improve it:
1) Table 2: This table is a little difficult to read. It may be helpful to add vertical lines so the columns are more distinct. Right now it is difficult to tell where the P and N belong.
2) Line 60-63: Please discuss why the results of animal studies and those of human studies are contradictory to each other, in particular the changes of Firmicutes and Proteobacteria.
3) The authors mentioned decreased intestinal permeability but avoided the use of "leaky gut." Why?
4) Not sure if Figure 1 is informative because dietary interventions are different among different reports.
5) Figure 2 has many formatting symbols that need to be removed.
Author Response
REVIEWER 1
Comment 1
Table 2: This table is a little difficult to read. It may be helpful to add vertical lines so the columns are more distinct. Right now it is difficult to tell where the P and N belong.
As the reviewer indicated, we added vertical lines in Table 2.
Page 13 line 397: Corrected Table 2 is inserted.
Comment 2
Line 60-63: Please discuss why the results of animal studies and those of human studies are contradictory to each other, in particular the changes of Firmicutes and Proteobacteria.
It has been reported that changes in the gut microbiota at the phylum levels (e.g., Firmicutes, Bacteroidetes, Proteobacteria) due to fat intake are not only influenced by the amount of fat ingested, but also by the quality of the fat. In animal experiments, a high-fat diet with lard is widely used (e.g., Research Diets Inc., catalog# D12451), but fat intake in human studies is implemented by commercial foods. Therefore, the difference in the quality of ingested fat is considered to be a factor in the difference of changes in the gut flora between animal and human studies. On the other hand, it has been suggested that the development of metabolic endotoxemia due to fat intake may involve the growth of certain bacteria. Therefore, in this paper, we consider the intestinal bacteria at the bacterial genus level that connect food factors and metabolic endotoxemia. Based on the above thoughts, we corrected the manuscript as below.
From Page 2 line 63 To Page 2 line 88: One possible cause of the different changes in the gut microbiota at the phylum level (e.g., Firmicutes, Bacteroidetes, Proteobacteria) in human and animal studies is the difference in the type of fat consumed. The high-fat diet used in animal experiments (e.g., Research Diets Inc., catalog# D12451) contains lard, while human studies assess fat intake in daily diets. Devkota et al. evaluated the gut microbiota in C57BL/6 mice fed a low-fat diet, a high-fat diet with lard, or a high-fat diet with milk fat for 21 days [12]. In this experiment, both high-fat diets were isocaloric, rich in saturated fatty acids, and 37% of the ingested kcal were from fat. As a result, the proportion of Firmicutes increased and that of Bacteroidetes decreased in the gut microbiota of mice fed a high-fat diet containing lard, compared to mice fed a low-fat diet. In contrast, in mice fed a high-fat diet containing milk fat, the proportion of Firmicutes decreased and that of Bacteroidetes increased compared to the low-fat diet fed mice. Interestingly, Devkota et al. also identified specific bacteria that increased only by ingestion of a high-fat diet containing milk fat [12]. Compared to mice fed a low-fat diet, or a high-fat diet containing lard, mice fed with a high-fat diet containing milk fat had increased proportions of Bilophila wadsworthia, a sulfite-reducing bacterium, in gut microbiota. They also elucidated the mechanism underlying this increase; intake of milk fat increased the level of taurocholic acid in bile. Bilophila wadsworthia populations increased by utilizing sulfur components in taurocholic acid, causing intestinal inflammation in mice. An increase in total fecal bile acid and a concomitant increase in Bilophila wadsworthia in the gut microbiota was also reported in humans upon dietary intake of animal fat [13]. Natividad et al. also showed that increased Bilophila wadsworthia in mice fed a high-fat diet contributed to increased blood LPS levels (they measured soluble CD14 as a surrogate marker), increased fasting blood glucose levels, and the development of a fatty liver [14]. As Helicobacter pylori was discovered as a pathogen in gastric cancer, some pathobionts may also exist for induction of metabolic endotoxemia (however, this cannot be detected by evaluating changes of the gut flora at the phylum levels). We further discuss the bacterial genera that are thought to be associated with metabolic endotoxemia in Section 4.”
Page 2 line 61: Following sentence was deleted because it is unnecessary information to consider the difference between the results of animal experiments and human tests; “and Defferibacteres”
Page 2 line 61: We corrected error as follows; From “Bacteroides” To “Bacteroidetes”
Page 2 line 62: Following sentence was deleted because it is unnecessary information to consider the difference between the results of animal experiments and human tests; “and Actinobacteria”
Page 2 line 62: We corrected error as follows; From “Bacteroides” To “Bacteroidetes”
Page 35 line 769: We appended the sentences so that it is in line with the argument as follows; “to find true pathobionts”
Comment 3
The authors mentioned decreased intestinal permeability but avoided the use of "leaky gut." Why?
As the reviewer has pointed out, the state of increased intestinal permeability is called as "leaky gut". However, in the papers cited in this manuscript, the expression "intestinal permeability" was often used. Therefore, we used in this manuscript the term "intestinal permeability" to unify the expression of it with the cited papers.
Comment 4
Not sure if Figure 1 is informative because dietary interventions are different among different reports.
As the reviewer has pointed out, different dietary factors that administered to animals have different effects on the gut microbiota. However, the outcome of intervention, the reduction of blood LPS (or LBP) levels, are commonly observed in these experiments. We believe that intestinal bacterial that changed commonly in these studies may be involved in metabolic endotoxemia, and thus we created Figure 1. The results in Figure 1 are used for the discussion in section 4.
Comment 5
Figure 2 has many formatting symbols that need to be removed.
We apologize for the inappropriate symbols in Figure 2. We corrected them as below.
Page 34 line 748: Corrected Figure 2 is inserted.
Reviewer 2 Report
This is an engaging review dealing with the role of fat intake in the development of metabolic endotoxemia. Authors also have provided a large amount of data concerning the capability of dietary factors in modulating this response by affecting the gut microbiota. However, the readability of the manuscript is not entirely satisfying, and the presentation of results is somewhat confusing. Therefore some issues, as outlined below, should be addressed.
Major comments:
1. The critical section “4. Association of dietary factor-induced reduction of blood LPS and modulation of the gut microbiota”, is rather poorly discussed. Authors should provide a clear link between changes in the gut microbiota presented in Table 4 and endpoints presented in Table 3 in the context of influence of dietary factors since the aim of this review was to “summarize the recent findings in humans and animals about dietary factors that attenuate metabolic endotoxemia, focusing on regulation of gut microbiota”. Also, there is a lack of data on the impact of Lactobacillus rhamnosus on changes in gut microbiota composition, although this bacteria, in Table 3, is presented to decrease levels of LPS and LBP [52,53].
2. Some data presented in tables, for example concerning the influence of vitamins, sulfated polysaccharide, extracts (including broccoli sprout extract) and foods on the blood LPS level, are not commented on at all.
3. The data presented in tables in columns “Improvement of the gut microbiota” and “Change of major gut microbe” are not clear.
4. The section “5. Perspective: “Sulforaphane”, the dietary factor expected to reduce LPS-producing bacteria Desulfovibrio”, seems to be detached from the rest of the manuscript since no data supporting the protective role of sulforaphane or broccoli, with the exception of only one study on the effects of broccoli sprout extract on LPS level in the blood (presented in Table 3) against an increase in blood LPS level, are discussed in the previous sections.
Minor comments
1. Please remove non-printed characters from Figure 2.
2. References to all tables should be provided in the main text.
3. In the legends of tables, please explain the meaning of different arrows.
Reviewer 3 Report
The authors attempt to review the dietary factors that attenuate metabolic endotoxemia, focusing on the regulation of gut microbiota is very good. Besides, correcting for some formatting and grammatical errors, I believe addressing the limitations/suggestions mentioned below will improve the manuscript.
Major Comments:
The authors should have a paragraph on current limitations, current advancements in microbiota tools and models and how those advancements will impact future studies.
Minor Comments:
Lines 63-65: please rewrite the sentence: ‘Furthermore, although not due…..human studies’. As it is confusing to understand what authors are trying to convey.
Lines 120-129: please crosscheck references 31 and 32 as there seems to be some confusion in the sentence ‘Therefore, the reason for the lack of increase in blood LPS….’. The authors initially mention that Anitha et al. found increased blood LPS level (Line 121) and then later wrote a lack of increase in blood LPS in the Anitha et al. study (Line 127)
Line 271: specify that subjects were calorie-restricted to 800 kcal/day but consumed the same normal diet.
Line 281: Table 1- specify the type of insoluble fiber used in Ref #63
Line 283: Table 1- to be consistent please specify the type of bacteria altered by fruits and vegetables consumption rather than mention it as change in alpha diversity Ref #14
Line 288: Table 3- to be consistent please specify the type of diet and the rat or mice strain used for the study (like done for Ref #52). Also replace ‘healthy mouse’ with actual mice strain and type of diet fed (for example C57BL/6J mice fed chow diet)
Line 302-304: specify the readout (mRNA abundance or protein levels) to clarify how the study (Ref# 111) reported enhanced intestinal tight junction.
Figures 1 and 2: remove the pilcrow sign
Author Response
REVIEWER 3
Major Comment 1
The authors should have a paragraph on current limitations, current advancements in microbiota tools and models and how those advancements will impact future studies.
Based on the reviewer’s comments, we added the sentence for the current issue of the gut microbiota analysis technology and future prospects, as below.
Page 35 line 775: We added the sentences as follows; “The development and popularization of next-generation sequencing has made it possible to comprehensively analyze the “fecal” microbiota in humans. On the other hand, as mentioned above, there are also mucin-adherent bacteria that are thought to be involved in metabolic endotoxemia (e.g., Akkermansia and Bacteroides). In a colitis mouse model, it has been reported that the bacterial flora in the mucin layer exhibits changes from 12 weeks before the onset of colitis, and that the mucin layer was thinned [170]. In this study, changes in the fecal flora occurred at the same time as the onset of colitis, indicating that the bacteria in the mucin layer play an important role in understanding the physiological state of the intestinal tract. However, although it is possible to collect mucin layer samples in animals, it is not easy to do so in humans, due to ethical and technical obstacles. In the future, if a method for collecting the mucin layer in a noninvasive manner is established in humans, the research field of metabolic endotoxemia can be further advanced.”
Minor Comment 1
Lines 63-65: please rewrite the sentence: ‘Furthermore, although not due…..human studies’. As it is confusing to understand what authors are trying to convey.
The sentence is not relevant to the argment of this manuscript, fat intake and dyslipidemia, and thus we deleted it.
From Page 2 line 88 To Page 3 line 100: The following sentences are deleted; “Furthermore, although not due to ingestion of a high-fat diet, fecal bacteria, that composition was associated with blood LPS levels, have been reported in several human studies. Dewlf et al. resported that Bacteroides positively correlated with blood LPS levels and that Firmicutes, Bacilli, Actinobacterium, Bifidobacterium, Faecalibacterium prausnitzii, Lactobacillus gasseri, and Eubacterium biform were negatively correlated [13]. Pedersen et al. reported that unclassified_Clostridia, unclassified_Actinobacteria, unclassified_Porphyromonadaceae, and Flavonifractor were positively correlated with blood LPS levels, while unclassified_Bacteroidals was negatively correlated [15]. Kopf et al. reported that Bacteroides was positively correlated with blood LPS binding protein (LBP) levels, while Firmicutes was negatively correlated [16]. Luthold et al. reported that Faecalibacterium was negatively correlated with blood LPS levels [17]. Thus, significant changes in the composition of the major enteric bacterial groups, Firmicutes and Bacteroides, have been reported in animals and humans in relation to blood LPS levels, but this is not necessarily consistent between species and between studies.”
Minor Comment 2
Lines 120-129: please crosscheck references 31 and 32 as there seems to be some confusion in the sentence ‘Therefore, the reason for the lack of increase in blood LPS….’. The authors initially mention that Anitha et al. found increased blood LPS level (Line 121) and then later wrote a lack of increase in blood LPS in the Anitha et al. study (Line 127)
We apologize for error in writing. The sentence is corrected as below.
Page 4 line 152: “Anitha” is corrected to “Reichardt”
Minor Comment 3
Line 271: specify that subjects were calorie-restricted to 800 kcal/day but consumed the same normal diet.
We corrected sentence as below based on the information in referenced paper.
Page 8 line 366: The sentence is corrected as follows; From “intake of a diet of 800 kcal/day” To “intake of a defined formula diet of 800 kcal/day”
Minor Comment 4
Line 281: Table 1- specify the type of insoluble fiber used in Ref #63
We corrected the description as below based on the information in referenced paper.
Page 10 line 376 (Table 1): The sentence is corrected as follows; From “Insoluble dietary fiber” To “Insoluble dietary fiber [from Fiber One Original cereal (General mills)]”
Minor Comment 5
Line 283: Table 1- to be consistent please specify the type of bacteria altered by fruits and vegetables consumption rather than mention it as change in alpha diversity Ref #14
In the paper, not only α-diversity of gut flora but also OUT analysis was performed, but it seemed that the proportion of bacteria did not change at the genus levels. I added that information in the revised manuscript as below.
Page 11 line 384 (Table 1): The sentence is corrected as follows; From “α-diversity” To “α-diversity (No significant change in bacterial genera was found)”
Minor Comment 6
Line 288: Table 3- to be consistent please specify the type of diet and the rat or mice strain used for the study (like done for Ref #52). Also replace ‘healthy mouse’ with actual mice strain and type of diet fed (for example C57BL/6J mice fed chow diet)
We describe the strains of laboratory animals and the type of food used in all the studies listed in Table 3.
Please refer to Table 3 (From Page 14 line 406 To Page 18 line 429) in corrected manuscript.
Minor Comment 7
Line 302-304: specify the readout (mRNA abundance or protein levels) to clarify how the study (Ref# 111) reported enhanced intestinal tight junction.
We added that information in the manuscript as below.
Page 19 line 434: The following sentence is added; “by preserving claudin 4 and occludin localization at tight junctions,”
Minor Comment 8
Figures 1 and 2: remove the pilcrow sign.
We apologize for the inappropriate symbols in Figure 1 and 2. We corrected them as below.
Page 32 line 700: Corrected Figure 1 is inserted.
Page 34 line 748: Corrected Figure 2 is inserted.
Round 2
Reviewer 2 Report
N/A